# Targeted Gene Next-Generation Sequencing Panel in Patients with Advanced Lung Adenocarcinoma: Paving the Way for Clinical Implementation

**DOI:** 10.3390/cancers11091229

**Published:** 2019-08-22

**Authors:** Maria Gabriela O. Fernandes, Maria Jacob, Natália Martins, Conceição Souto Moura, Susana Guimarães, Joana Pereira Reis, Ana Justino, Maria João Pina, Luís Cirnes, Catarina Sousa, Josué Pinto, José Agostinho Marques, José Carlos Machado, Venceslau Hespanhol, José Luis Costa

**Affiliations:** 1Pulmonology Department, Centro Hospitalar Universitário de São João, Alameda Prof. Hernani Monteiro, 4200-319 Porto, Portugal; 2Faculty of Medicine, University of Porto, Alameda Prof. Hernani Monteiro, 4200-319 Porto, Portugal; 3Institute of Molecular Pathology and Immunology of the University of Porto (IPATIMUP), 4200-135 Porto, Portugal; 4Institute for Research and Innovation in Health (i3S), University of Porto, Rua Alfredo Allen, 4200-135 Porto, Portugal; 5Pathology Department, Centro Hospitalar Universitário de São João, Alameda Prof. Hernani Monteiro, 4200-319 Porto, Portugal; 6Escola Superior de Saúde (ESS), Instituto Politecnico do Porto (IPP), Rua Dr António Bernardino de Almeida, 4200-072 Porto, Portugal

**Keywords:** lung cancer, targeted therapy, next-generation sequencing, molecular profiling

## Abstract

Identification of targetable molecular changes is essential for selecting appropriate treatment in patients with advanced lung adenocarcinoma. *Methods*: In this study, a Sanger sequencing plus Fluorescence In Situ Hybridization (FISH) sequential approach was compared with a Next-Generation Sequencing (NGS)-based approach for the detection of actionable genomic mutations in an experimental cohort (EC) of 117 patients with advanced lung adenocarcinoma. Its applicability was assessed in small biopsies and cytology specimens previously tested for epidermal growth factor receptor (EGFR) and anaplastic lymphoma kinase (ALK) mutational status, comparing the molecular changes identified and the impact on clinical outcomes. Subsequently, an NGS-based approach was applied and tested in an implementation cohort (IC) in clinical practice. Using Sanger and FISH, patients were classified as EGFR-mutated (*n* = 22, 18.8%), ALK-mutated (*n* = 9, 7.7%), and unclassifiable (UC) (*n* = 86, 73.5%). Retesting the EC with NGS led to the identification of at least one gene variant in 56 (47.9%) patients, totaling 68 variants among all samples. Still, in the EC, combining NGS plus FISH for ALK, patients were classified as 23 (19.7%) EGFR; 20 (17.1%) KRAS; five (4.3%) B-Raf proto-oncogene (BRAF); one (0.9%) Erb-B2 Receptor Tyrosine Kinase 2 (ERBB2); one (0.9%) STK11; one (0.9%) TP53, and nine (7.7%) ALK mutated. Only 57 (48.7%) remained genomically UC, reducing the UC rate by 24.8%. Fourteen (12.0%) patients presented synchronous alterations. Concordance between NGS and Sanger for EGFR status was very high (κ = 0.972; 99.1%). In the IC, a combined DNA and RNA NGS panel was used in 123 patients. Genomic variants were found in 79 (64.2%). In addition, eight (6.3%) EML4-ALK, four (3.1%), KIF5B-RET, four (3.1%) CD74-ROS1, one (0.8%) TPM3-NTRK translocations and three (2.4%) exon 14 skipping MET Proto-Oncogene (MET) mutations were detected, and 36% were treatable alterations. *Conclusions*: This study supports the use of NGS as the first-line test for genomic profiling of patients with advanced lung adenocarcinoma.

## 1. Introduction

Lung cancer is the most common cause of cancer-related death worldwide [1], being frequently diagnosed in an advanced stage when curative treatment is no longer possible. Histologically, lung cancer is divided into small and non-small cell lung cancer (NSCLC). NSCLC is the most common type and includes squamous cell carcinoma, large-cell carcinoma and adenocarcinoma. Adenocarcinoma subtype accounts for more than half and is further defined according to different molecular subtypes by the identification of oncogenic drivers [2].

Over the last few decades, conventional platinum-based chemotherapy has produced only a modest increase in patient´s overall survival, reaching a plateau, with response rates around 35% and median survival time of 10–12 months [3]. Recent advances in the knowledge of NSCLC biology, especially on the discovery of oncogenic molecular changes leading to aberrant activation of intracellular signalling associated with the sustained growth of lung cancer cells, led to the development of genotype-targeted therapies, with significant improvement in patient´s outcomes. For instance, the identification of epidermal growth factor receptor (EGFR) kinase domain activating mutations, conferring sensitivity to EGFR tyrosine kinases inhibitors, changed the lung cancer treatment paradigm and contributed to increased progression-free survival (PFS) and quality of life in this subgroup of patients [4]. Targeted therapy for patients with anaplastic lymphoma kinase (ALK) translocations had a similar impact on prognosis [5]. In addition, the identification of other rare genomic alterations involving the ROS Proto-Oncogene 1 (ROS1), B-Raf proto-oncogene (BRAF), MET Proto-Oncogene (MET), Erb-B2 Receptor Tyrosine Kinase 2 (ERBB2), and ret proto-oncogene (RET) genes led to the development of new targeted therapies, some already approved for first-line use [2,3,4,5,6]. Additionally, genomically unclassifiable (UC) patients are candidates for immunotherapy with checkpoint inhibitors [6].

Updated recommendations from the College of American Pathologists, International Association for the Study of Lung Cancer, Association for Molecular Pathology (CAP/IASCL/AMP), and European Society of Medical Oncology (ESMO) have strengthened the 2013 guidelines and suggest genomic testing for EGFR, ALK, and ROS1 for all advanced NSCLC, regardless of patients’ characteristics [7]. Besides, there is strong advise to perform broader molecular profiling for detection of rare mutations, to which targeted therapies are available or suitable for off-label treatment or clinical trials (ERBB2, MET, BRAF, KRAS, and RET) [8,9]. 

In the setting of lung cancer advanced disease, molecular diagnosis faces several difficulties. Foremost, both the quantity and quality of tumor tissue and nucleic acid content are critical. Tissue samples are often small histological fragments or cytological specimens obtained by bronchoscopy or transthoracic biopsy. These samples should be analyzed for histopathology, which also includes the programmed death-ligand 1 (PD-L1) test for immunotherapy selection and molecular testing [9]. The minimal conventional genomic study generally includes, sequentially or in parallel, at least, the EGFR and ALK mutational analysis. Sequential determinations require a substantial DNA amount, are time-consuming, cause sample´s exhaustion and often leads to under genotyping and treatment delay [10,11].

Next-Generation Sequencing (NGS) allows the sequencing of several genomic regions in a single test, in a single platform and even in samples with low DNA content. There are different NGS tests for whole genome, whole exome, or selected genes, depending on the application purpose. An NGS-based approach potentially provides a more sensitive and comprehensive genetic characterisation of lung cancer, which may impact the therapeutic options and patient´s prognosis. 

In this study, we tested an NGS-based approach for the detection of actionable genomic mutations in a cohort of patients with advanced lung adenocarcinoma, previously tested sequentially for EGFR and ALK mutational status, and evaluated the clinical impact of the NGS-based approach. The primary endpoint was to assess the added-value of the NGS strategy over the sequential approach and the concordance regarding the EGFR status. Additionally, the results from the clinical implementation of a combined NGS DNA plus RNA panel are presented.

## 2. Material and Methods

### 2.1. Study Design

A total of 240 patients with advanced lung adenocarcinoma, diagnosed in the Pulmonology Department of Centro Hospitalar Universitário de São João (CHUSJ), EPE were enrolled in this study and divided into two groups: (i) experimental cohort (EC), corresponding to patients diagnosed between January 2015 and December 2016. Of a total of 127 patients previously examined for EGFR and ALK mutational status, 117 had tumor samples available for NGS retest. Ten patients were excluded from the study, seven owing to insufficient tumor sample, two with stage IIIA, and one included in another trial; (ii) implementation cohort (IC), corresponded to 123 patients diagnosed from September 2017 to July 2018, to whom NGS was integrated into the daily clinical practice (Figure 1). 

The tumor staging was based on the 7th edition of the TNM staging system until December 2017 and the 8th edition from January 2018 [12,13].

All subjects gave their informed consent for inclusion before they participated in the study. The study was conducted in accordance with the Declaration of Helsinki, and the protocol was approved by the Ethics Committee of CHUSJ (CES-108/14).

### 2.2. Tumor Specimens 

Biopsy and cytology specimens, from the primary tumor and metastatic sites, were reviewed by pathologists. Histological specimens were fixed with formalin (formalin-fixed paraffin-embedded tissue, FFPE) and cytological specimens as smears or cellblocks. After pathological and immunohistochemical evaluation, samples were used for DNA extraction using the QIAamp DNA Mini Kit (Qiagen, Hilden, Germany), following manufacturer´s instructions. DNA was quantified with NanoDrop Lite Spectrophotometer (Thermo Fisher Scientific™, Waltham, MA, USA) or Qubit^®^ 2.0 Fluorometer (Invitrogen, Waltham, MA, USA). All genetic analyses were done at IPATIMUP, a College of American Pathologists and ISO15189 accredited laboratory.

### 2.3. Library and Template Preparation for Next-Generation Sequencing

The Ion AmpliSeq Colon and Lung Cancer Research Panel v2 (Ion Torrent, Waltham, MA, USA) was used to detect changes in DNA, in the experimental phase. This multiplex PCR-based test allows the analysis of 1850 hotspots and targeted regions in 22 genes (*AKT1*, *ALK*, *BRAF*, *CTNNB1*, *DDR2*, *EGFR*, *ERBB2*, *ERBB4*, *FBX7*, *FGFR3*, *FGFR1*, *FGFR2*, *KRAS*, *MAP2K1*, *MET*, *NOTCH1*, *NRAS*, *PTEN*, *PIK3CA*, *STK11*, *SMAD4* and *TP53*) involved in tumorigenesis. Libraries were generated using 1–10 ng of DNA from tissue FFPE blocks sections, according to the manufacturer. 

In the clinical implementation phase, samples were characterised using the Oncomine Solid Tumor kits CE-IVD (Ion Torrent, Waltham, MA, USA). These assays allow the analysis of the same targets as the Ion AmpliSeq Colon and Lung Cancer Research Panel v2 plus the detection of *ALK*, *RET*, *ROS1*, and *NTRK1* gene fusions. Libraries were generated using 1–10 ng of DNA and RNA from tissue FFPE blocks sections, according to the manufacturer. The final libraries were quantified by qPCR with the Ion Library TaqMan^®^ Quantitation Kit (Ion Torrent, Waltham, MA, USA) and used for template preparation performed using the Ion Chef (Ion Torrent, Waltham, MA, USA).

### 2.4. Next-Generation Sequencing and Bioinformatic Analysis

Loaded chips were sequenced in an Ion PGM or Ion S5XL sequencer. Sequencing quality was assessed through the plug-in coverage analysis and the samples were analyzed using dedicated bioinformatic workflows within the Ion Reporter v5.6 server (Ion Torrent, Waltham, MA, USA). Samples with a number of reads <100,000 and/or the average base coverage <500× were considered inadequate for analysis. The amplicons with a coverage <250× were considered non-informative. Mutations with allele frequencies of at least 10% and adequate coverage in target regions were considered to call a mutation in a patient sample. Polymorphisms, synonymous or intronic mutations were excluded. The Catalogue of Somatic Mutations in Cancer (COSMIC) database was used to access the clinically relevant variants. Some mutations detected by NGS were validated by digital PCR. 

### 2.5. Statistical Analysis

The sample size was limited by the availability of specimens for subsequent NGS analysis following sequential standard molecular diagnostic approach. Most of the analysis was descriptive. Categorical data were described as absolute (*n*) and relative frequencies. Medians, interquartile ranges (IQR), and minimum and maximum values were determined for continuous variables. The NGS results were correlated with other parameters and assessed with the chi-square test or Fisher´s exact test, when appropriate. Cohen´s Kappa test was used to assess the inter-rater agreement for categorical data. Kaplan-Meier actuarial curve analysis was used to assess survival and the log-rank test for the chi-square (X^2^) calculus, for each event time and each evaluated group. The significance level assumed was 0.05. All statistical analyses were performed using the Statistical Package for Social Sciences (SPSS, IBM Corp, Chicago, IL, USA) software, version 25.0.

## 3. Results

### 3.1. Clinicopathologic Characteristics and Mutational Profile

In this study, 117 and 123 patients were, respectively, included in the EC and IC groups. The clinicopathologic features are presented in Table 1. All specimens were adenocarcinoma, except one suggestive of adenosquamous carcinoma (in the EC). Most of patients had stage IV disease at the time of diagnosis. In both groups, samples were predominantly core biopsies suitable for histological analysis. Cytological samples were obtained by fine-needle-aspiration techniques, as pleural and pericardial fluid aspiration, endobronchial ultrasound-guided needle aspiration (EBUS-TBNA), and lung and peripheral lymph nodes fine-needle aspirations (Table 1).

NGS analysis succeeded in both histological and cytological samples, with no statistically significant differences between groups (*p* = 0.487). The frequency of alterations found did not differ in both histological and cytological samples between both cohorts. In the EC, 7/21 mutations were from cytological samples and 49/96 in histological samples (*p* = 0.141). In the IC, hotspot alterations were found in 12/18 cytological and 66/105 histological samples (*p* = 0.538).

In the EC, the *EGFR* mutation test was performed in all patients, with a positive rate of 18.8% (22/117). *ALK* translocations were identified in 7.7% of cases (9/117) (Table 1). The remaining cases were designated as UC (*n* = 86, 73.5%). Of the 95 *EGFR* negative cases with *ALK* FISH testing indication, 17.9% (17/95) had insufficient sample for *ALK* test. *EGFR*-mutated patients were distributed as follows: 16 (72.7%) exon19 deletions (19 DEL), five (22.7%) L858R, and one (4.5%) exon20 insertion (20 Ins) (Appendix A).

### 3.2. Next-Generation Sequencing Results

According to the results obtained with the Ion AmpliSeq^TM^ Colon and Lung Cancer Research Panel v2 for DNA analysis, 56 (47.9%) patients harbored at least one gene variant and in 61 (52.1%) no genomic alteration was identifiable, remaining as UC. Analyzing by patient and comparing with the standard approach (Figure 2A), the most frequent alterations were *KRAS* (*n* = 23, 19.7%), *EGFR* (*n* = 23, 19.7%), *BRAF* (*n* = 5, 4.3%), TP53 (*n* = 3, 2.6%), *ERBB2* (*n* = 1, 0.9%), and *STK11* (*n* = 1, 0.9%). Analyzing by sample, a total of 68 genomic variants were identified (Figure 2B; Appendix A).

Of the 86 (73.5%) patients classified as UC by the standard approach, 29 (33.7%) had at least one gene alteration. Beyond *EGFR* mutations, variants were observed in other targetable genes, as described above. Two additional *EGFR* mutations were found, one not previously identified and one as co-alteration.

Combining NGS for DNA analysis data plus ALK detection by FISH, 60 (51.3%) patients had at least one alteration identified, and 57 (48.7%) remained without identified mutations, reducing the UC rate by 24.8% (Figure 3A,B).

### 3.3. Concurrent Genomic Alterations

Concurrent changes were found in 14 patients (12.0%) (Table 2), with emphasis on co-alterations occurring in *EGFR* (4/23) and ALK (5/9) mutated patients. Among the *EGFR* mutated patients, one exhibited an *ALK* point mutation c.3512T>A in a neglectable percentage; one patient, with a combination of Del19 with allelic frequency of 66% plus p.T790M with allelic frequency of 0.6%, had a PFS with a 1st generation TKI of 9.2 months and progressed with the p.T790M point mutation; the other two patients had a *KRAS* mutation and a *PIK3CA* mutation, respectively. In the ALK-positive patients’ subgroup, a molecular co-alteration was detected in five cases: three with *KRAS* and two with *TP53* mutations. Two of the ALK plus *KRAS* mutated patients had a very dismal evolution with overall survival (OS) less than 3 months (Table 2).

### 3.4. Concordance Between Sanger and NGS for the EGFR Status

A set of 117 patients was analyzed by both Sanger sequencing and NGS for *EGFR* mutations detection, allowing the concordance determination between tests. Overall, the percentage of concordant cases was 116/117 (99.1%) (Table 3), and there was an almost perfect agreement between the two tests (κ = 0.972; *p* < 0.001).

Assuming that all *EGFR* positive cases are true positives, in this cohort, 23 patients were *EGFR* mutated. The sensitivity for Sanger was 95.6% and for NGS 100%. One extra *EGFR* mutated patient was found among the unclassified population by Sanger.

### 3.5. Impact of the Identification of Targetable Alterations in Patient’s Overall Survival

The EC presented a median OS of 13.0 months (95% CI 8.1–17.9, sd = 2.5): 24 months (95% CI 10.2–37.8, sd = 7.0) for the EGFR subgroup, 11 months (95% CI 5.2–16.8, sd = 2.98) for ALK, and 11 months for UC patients (95% CI 5.98–16.0, sd = 2.6). Specific survival patterns were observed in subtypes of oncogenic alterations detected by NGS, with EGFR patients having the best median OS and KRAS the worse (Figure 4A). Among patients with a druggable oncogenic alteration, 30 were treated with targeted therapy, 21/23 *EGFR*, 2/5 *BRAF*, 7/9 (*ALK*) were treated with TKIs. Regardless of the treatment line, OS was significantly higher for this subgroup of patients (median OS 24 vs. 9 months, *p* = 0.028) (Figure 4B). As groups were small, no statistical comparisons were feasible.

### 3.6. NGS Results from the Implementation Cohort with a Combined DNA and RNA Panel

Combined DNA and RNA sequencing succeeded in 117 (95.1%). DNA degradation occurred in four (3.2%) cases and in these cases, *EGFR* was analyzed by Sanger sequencing. RNA extraction failed in two cases (1.6%), with no statistically significant differences in the sequence rate according to the sample type (16/18 for cytology and 102/105 for histology, *p* = 0.1).

DNA and RNA hotspot alterations were detected in 79 (64.2%) patients. *KRAS* mutations occurred in 33 (26.8%) patients, *EGFR* in 18 (14.6%), *BRAF* in four (3.2%), *ERBB2* in five (4.1%), and *PIK3CA* in one (0.8%). Exon 14 skipping *MET* mutation was detected in three (2.4%) patients. Considering gene translocations, eight (6.5%) patients had an ELM4-ALK, four (3.1%) an KIF5B-RET, one (0.8%) an CD74-ROS1, and one (0.8%) an TPM3-NTRK (Figure 3C and Appendix A). Only 44 (35.8%) patients remained as UC. Forty-four (35.8%) patients had treatable alterations. Additionally, co-alterations were found in 11 (8.9%) cases (Appendix A).

## 4. Discussion

Lung cancer treatment is rapidly evolving from histology to precision-based therapy, relying on predictive biomarkers determined by molecular tests. Sequential analysis of each gene with single-gene assays is unable to search efficiently for all actionable mutations, due to tumor sample extinction. For completing the minimum genomic characterisation, patients would have to undergo additional diagnostic procedures, with increased risk and costs. Testing each biomarker, one at a time, may result in longer and potentially unacceptable turnaround time. Besides, and considering the number of genes that must be tested, it will probably not be cost-effective. 

Lung cancer diagnosis is established predominantly in small biopsies or cytological samples, which must be enough for morphology, immunohistochemistry (IHC), and molecular evaluation, essential to determine if an actionable oncogenic alteration is present. In real-world clinical practice, many obstacles arise to meet this need. The barriers consist of cost, number of separate assessments required, quality of tumor samples and local availability of the tests [14,15].

In the last years, Sanger sequencing has been considered the gold standard for gene mutation testing, particularly, for EGFR. However, this technique is not very sensitive, requires samples with at least 20% of tumor cells and its application to small lung biopsies or cytology specimens is not always possible. In addition, to assess ALK rearrangements, the gold standard is IHC or FISH, both tests needing many histological sections from an FFPE or a cellblock. 

High-throughput NGS, also referred to as massively parallel sequencing, allows multiplex PCR with simultaneous amplification of a pre-specified panel of genes in a single reaction. Depending on the panel, it offers the possibility to analyze DNA, RNA, transcription regions, methylation patterns and, more recently, tumor mutation burden. Targeted gene panels represent an alternative method for capturing specific genomic regions for subsequent sequencing and is the best option for molecular characterization of lung cancer, allowing multiple genes to be analyzed at the same time with enough depth of coverage to detect minor allele frequencies. Most NGS assays require as little as 10 ng of DNA, or even less, while non-NGS tests, like Sanger, require more DNA. NGS was previously validated in several studies and its superiority is recognised in terms of sensitivity, speed, and costs [16,17,18,19].

In the present study, we report our experience with a targeted NGS-based strategy, using the Thermofisher Ion Torrent NGS technology, with the Ion AmpliSeq Colon and Lung Cancer Research Panel v2, for the detection of actionable genomic alterations in a cohort of patients with lung adenocarcinoma, and the steps taken until its prospective integration into our clinical practice. This panel was previously validated in a study, in which the same sample was tested in seven different laboratories of the OncoNetwork Consortium [20]. In our study, tumor specimens were previously analyzed for the determination of EGFR and ALK mutational status, by Sanger sequencing and FISH, respectively, according to recommendations at that time [7]. After morphological and molecular standard subtyping, samples were retested with the targeted-gene NGS panel. NGS was done in samples with an average of 5.1 ng DNA, suggesting that successful sequencing may occur in samples with small DNA amounts, overcoming the difficulty of mutation analysis in the context of lung cancer diagnosis [15].

In our cohort, only 77 (65.8%) patients had both *EGFR* and *ALK* genes analyzed, reflecting the difficulty in completing the minimal recommended genomic analysis, based on single sequential tests, as previously mentioned. The specimens were diverse and the Ion Ampliseq sequencing proved to be effective even in critical samples, such as those obtained by FNA and biological fluids. Scarpa et al. [21] demonstrated, for the first time, the successful application of the Ion AmpliSeq Colon and Lung Cancer in cytological samples of lung adenocarcinoma, and these results were replicated in other studies [14,22,23,24]. 

NGS revealed excellent accuracy for *EGFR* detection and a very high concordance rate (99.1%) with Sanger sequencing. There was only one discordant case, an *EGFR* deletion (*EGFR* c.2236_2250del15) detected with NGS but missed by Sanger. Indeed, previously published studies have also highlighted the NGS superiority over the traditional methods [25,26,27]. Although not an epidemiological study, mutation patterns and frequencies agree with previous data published on our population [28].

Our data demonstrated high sensitivity to detect druggable alterations, significantly reducing the rate of genomically unclassifiable patients to 48.7%, with the combination of the Ion Ampliseq panel and ALK FISH test. Additionally, we identified molecular changes in 29/86 (33.7%) of the UC patients classified by the standard approach, as reported in similar studies [29,30]. Beyond KRAS mutations, one more patient with an *EGFR* activating mutation and five with BRAF-p.V600E could have been treated with targeted agents or included in clinical trials. 

NGS also provided some insights into the biology and clinical behavior of the disease. According to the survival analysis, subsets of patients could be defined: EGFR-mutated patients presented higher median OS and KRAS the lowest ones, which is similar to the literature data [31,32]. Unexpectedly, ALK patients had a low OS, perhaps due to the unavailability of TKIs as first-line treatment at the beginning of the study and to the coexistence of KRAS mutation in three of the nine *ALK* patients. This occurrence is rare, but it has already been reported as being associated with primary resistance to TKIs [32,33]. The expected co-occurrence of TP53 mutations with KRAS, EGFR, *BRAF*, *PIK3CA*, and *STK11* were also identified. Another interesting finding was the detection, before treatment, of a low allelic frequency p.T790M point mutation alongside the driver Del19, which, in this case, prompted progression through this on-target resistance mechanism. The identification of simultaneous alterations can explain the unexpected clinical outcomes and resistance patterns, suggesting the existence of subclonal populations or hypothetically a different population of tumor cells. Considering the small sample size of this study, definitive conclusions on the clinical impact of a concurrent genomic alteration cannot be driven and deserve further investigation.

After the evaluation phase of this DNA panel, since it has limitations regarding chromosomal rearrangements detection, a combined DNA and RNA panel, Ion Torrent^TM^ Oncomine™ Focus Assay [34], was integrated into the lung cancer molecular diagnosis. This panel allowed even a higher reduction of unclassified patients to 36%. In this clinical cohort, beyond the target point mutations and deletions found, exon 14 skipping MET mutation and gene translocations, ELM4-ALK, KIF5B-RET, CD74-ROS1, and even a TPM3-NTRK were detected. Overall, in the clinical cohort, and considering drug’s availability, about 36% of patients were candidates for specific therapy. These data demonstrate that this combined targeted NGS sequencing methodology allows the detection of common and some clinically relevant rare mutations. This strategy can identify more patients who are likely to benefit from specific therapies, improving clinical outcomes, minimising the toxic effects, and promoting drug development.

Our data are equivalent to those obtained in larger studies. In the Lung Cancer Consortium, an actionable oncogenic alteration was found in 64% of patients [30]. In a study made by Foundation Medicine, 6.832 clinical NSCLC samples were tested with a comprehensive hybridization capture panel with more than 300 genes, able to detect also translocations, and 71% of patients harbored at least one genomic alteration [35].

This study was practice changing. The routine diagnostic workup is, at present, a reflex test including IHC, PDL1, and a combined DNA and RNA next-generation sequencing on FFPE tissue or cytological samples, for all patients with NSCLC with genomic testing indication. This analysis was mainly focused on the clinical usefulness of NGS, in search of suitable patients for targeted approved drugs, off-label treatments and clinical trials, but the spectrum of applications goes further beyond that. This kind of combined panel, profiling DNA and RNA, has also the potential to detect alterations that may soon have available drugs and the ability to identify and further characterize resistance mechanisms and aberrant clinical patterns. Future challenges will be the interpretation of co-alterations, low-frequency variants, and those of uncertain biological meaning.

## 5. Conclusions

This study contributed to the clinical validation of NGS for screening actionable mutations in patients with lung adenocarcinoma, demonstrating good accuracy in biological specimens with small DNA content, like those obtained for lung cancer diagnosis. In addition, it provided a more comprehensive genomic characterisation of the disease, contributing to a better definition of tumor biology.

## Figures and Tables

**Figure 1 cancers-11-01229-f001:**
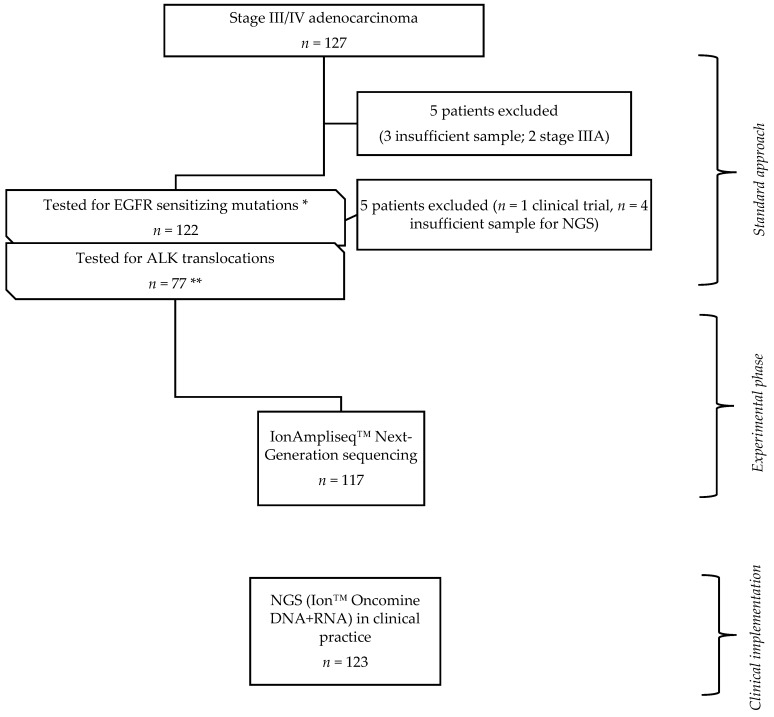
Study design; 127 patients genomically profiled with sanger and anaplastic lymphoma kinase (ALK) FISH test (standard approach) were selected. Among those, in 117 the same tumor sample previously tested was submitted to Next-Generation Sequencing (NGS), comprising the experimental phase of the study (experimental cohort (EC)). Among the EC, * 22/117 were epidermal growth factor receptor (EGFR) mutated and 95 had indication for anaplastic lymphoma kinase (ALK) testing, ** 17/95 patients did not perform it due to insufficient sample. After the EC, a combined DNA + RNA panel was applied to characterize genomically patients with lung adenocarcinoma; 123 cases were included for the purpose of this study (clinical implementation phase).

**Figure 2 cancers-11-01229-f002:**
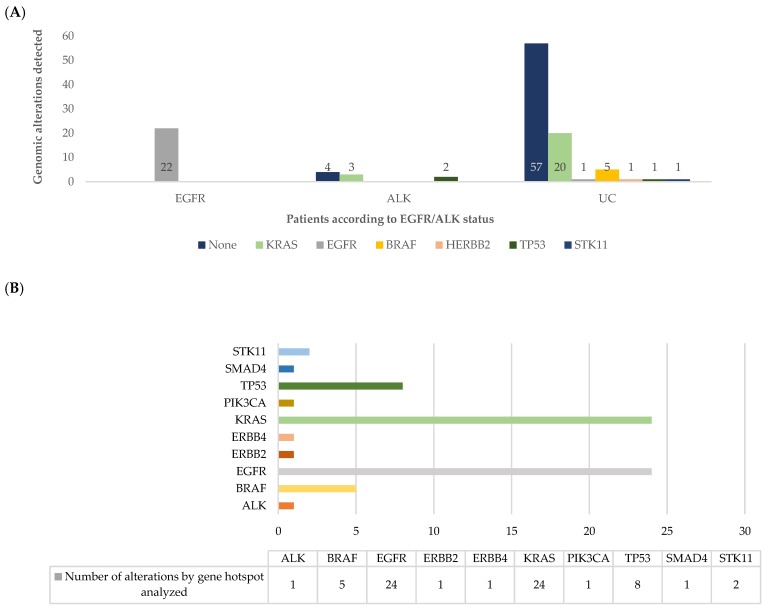
(**A**) Number of patients with genomic alterations assessed by NGS findings, in each patient´s group assessed by the standard classification (EGFR, ALK, and unclassifiable (UC)); (**B**) number of hotspot alterations in each gene analyzed by the Ion AmpliSeq™ assay.

**Figure 3 cancers-11-01229-f003:**
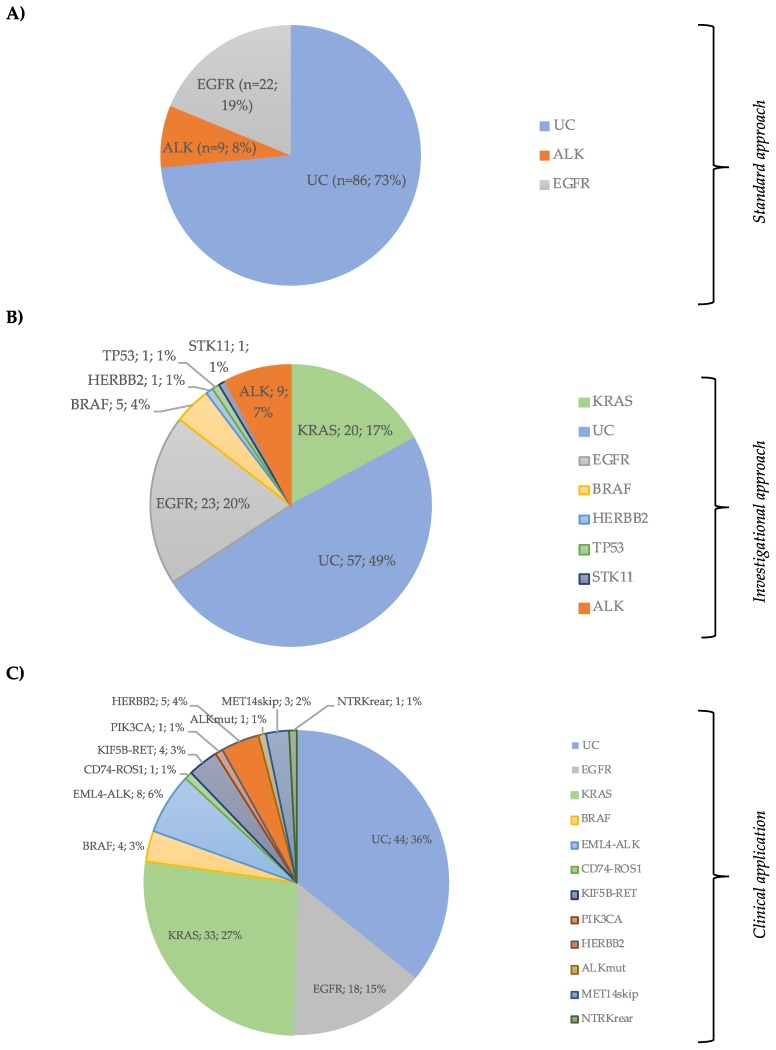
(**A**) Patient’s distribution by EGFR and ALK status (Standard Approach); (**B**) patient’s classification by Ion Ampliseq Lung and colon panel plus ALK FISH status (Investigational Approach); (**C**) distribution of hotspot DNA and RNA subgroups in the clinical cohort (Clinical application of the combined DNA + RNA Panel).

**Figure 4 cancers-11-01229-f004:**
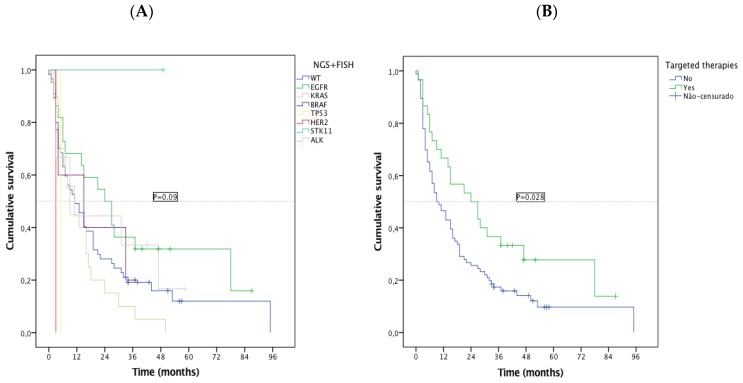
Kaplan-Meier overall survival (OS) by (**A**) NGS subgroups; (**B**) targeted treatment vs. non-targeted treatment.

**Table 1 cancers-11-01229-t001:** Patient´s demographics and clinical characteristics.

Characteristics				EC Value (*n*, %)	IC Value (*n*, %)	*p* Value
Age (median, range)				66 (38, 92)	67 (41, 94)	0.320
Gender	Male			71 (60.7)	77 (62.6)	0.760
	Female			46 (39.3)	46 (37.4)	
Performance status	0			42 (35.9)	61 (49.6)	0.055
	1			53 (45.3)	38 (30.9)	
	2			16 (13.7)	13 (10.6)	
	3			6 (5.1)	11 (8.9)	
Smoking status	Smoker/Former Smoker			74 (63.2)	82 (66.7)	0.579
	Never smoker			43 (36.8)	41 (33.3)	
Disease stage	III (A/B/C)			22 (18.8)	23 (18.7)	0.984
	IV			95 (81.2)	100 (81.3)	
Histology	Adenocarcinoma			116 (99.1)	123 (100)	0.304
	Adenosquamous			1 (0.9)	0	
TTF1 IHQ	Positive			101 (86.3)	100 (81.3)	0.070
	Negative			5 (4.3)	15 (12.2)	
	Unknown			11 (9.4)	8 (6.5)	
Specimen type	Histologic	Bronchial		31 (26.5)	22 (17.8)	0.487
		Lung		51 (43.6)	63 (51.2)	
		Pleura		10 (8.5)	7 (5.7)	
		Brain		2 (1.7)	5 (4.1)	
		Bone		2 (1.7)	1 (0.8)	
		Liver		0	2 (1.6)	
		Lymph node		0	3 (2.4)	
		Skin		0	1 (0.8)	
		Small bowel		0	1 (0.8)	
		Total		96 (82.1)	105 (85.4)	
	Cytologic	Lung-FNA		2 (1.7)	0	
		EBUS-FNA		2 (1.7)	2 (1.6)	
		Pleural fluid		12 (10.2)	10 (8.1)	
		Pericardial fluid		1 (0.9)	0	
		Bronchial washing or brushing		0	4 (3.2)	
		Lymph node		4 (3.4)	2 (1.6)	
		Total		21 (17.9)	18 (14.6)	
Molecular status (Standard approach)	EGFR	Mutated	22 (18.8)			
	ALK	EML4-ALK	9 (7.7)			
	UC		86 (73.5)			
	Total		117 (100)			

FNA, Fine-needle aspiration; UC, unclassifiable.

**Table 2 cancers-11-01229-t002:** Cases with concomitant genomic alterations.

Patient’s Classification (Standard Approach)	NGS Genomic Alteration	AF	Clinical Significance	PFS1	OS
UC	KRAS c.35G > T	26.7	Pathogenic	7.1	9
	TP53 c.839G > A	13.0	Likely pathogenic		
UC	BRAF c.1799T > A	50.4	Pathogenic	4	15
	TP53 c.476C > G	39.2	Uncertain		
UC	KRAS c.182A > G	8.8	Pathogenic	2	5
	TP53 c.461G > T	24.7	Uncertain		
	STK11p.Glu199Asp	12.8	Pathogenic		
UC	ERBB2 c.2310_2311insGCATAC	20	Not found	0	3
	TP53 c.1024C > T	21	Pathogenic		
UC	KRAS c.38_39delGCinsAA	14.2	n/a	0	4
	ERBB4 c.1033G > T	14.9	n/a		
EGFR	EGFR c.2235_2249del15	34.6	Pathogenic	10.1	37 (NR)
	PIK3CA c.1633G > A	27.8	Pathogenic		
EGFR	EGFR c.2236_2250del15	11.1	Pathogenic	8	24
	KRAS c.182A > G	0.38	Pathogenic		
EGFR	EGFR c.2240_2257del18	67.2	Pathogenic	9.4	40 (NR)
	EGFR c.2369C > T	0.6	Pathogenic		
EGFR	EGFR c.2239_2248del	66	Pathogenic	16.6	NR
	ALK c.3512T > A	0.08	Pathogenic		
EML4-ALK	TP53 c.538G > T	37.8	Pathogenic	9.7	42
EML4-ALK	TP53 c.524G > A	2.13	Pathogenic	16	31
EML4-ALK	KRAS c.35G > T	27.4	Pathogenic	NE	3
EML4-ALK	KRAS c.35G > T	11.7	Pathogenic	7	47
EML4-ALK	KRAS c.35G > A	6.1	Pathogenic	NE	2

UC, unclassifiable; AF, allelic frequency; PFS1, Progression Free Survival related to 1st Line therapy; OS, Overall survival, NE, not evaluated; NR, not reached: n/a, not available.

**Table 3 cancers-11-01229-t003:** Comparison between NGS and Sanger for EGFR status.

Gene	Cases Compared (*n*)	Concordant Results SS vs. NGS		Discordant SS vs. NGS		Concordant Cases (%)	Kappa
		Neg/Neg	Pos/Pos	Neg/Pos	Pos/Neg		
EGFR	117	94	22	1	0	99.1	0.972

NGS, next-generation sequencing; SS, Sanger sequencing.

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
