# Peer review of "Targeted Gene Next-Generation Sequencing Panel in Patients with Advanced Lung Adenocarcinoma: Paving the Way for Clinical Implementation"

_cancers, 2019, doi:10.3390/cancers11091229_

Round 1

Reviewer 1 Report

It's a very useful paper, especially for clinical detection. However, I have the following concerns:

1. The NGS for clinical validation is a good way, but how can you be sure that the results of NGS are accurate? Not a false positive?

2. For lung cancer, When NGS detects these high-frequency mutant genes, such as KRAS, EGFR, BRAF, TP53, HERBB2, and STK11. Is it possible to design primers for these high-frequency mutant genes, and only need to detect these genes without using NGS?

3 The raw data will open source?

Author Response

The NGS for clinical validation is a good way, but how can you be sure that the results of NGS are accurate? Not a false positive?

In order to minimize false positive and false negative results, stringent quality control metrics were applied to the data analysis. The minimum read depth and minimum percentage of mutated reads were set to eliminate random sequencing errors and provide high quality results. Using this settings we obtained >99% concordance between the different methodologies confirming the quality of the NGS results. Additionally, all NGS experiments were performed in a CAP and ISO15198 accredited laboratory.

For lung cancer, when NGS detects these high-frequency mutant genes, such as KRAS, EGFR, BRAF, TP53, HERBB2, and STK11. Is it possible to design primers for these high-frequency mutant genes, and only need to detect these genes without using NGS?

Thank you for the comment. Indeed, it would be possible to design primers for the frequently mutated genes and perform Sanger sequencing. This has been the standard strategy in past years. However, this has several limitations related with time, sample and cost that only a comprehensive strategy, like NGS, can address. In the context of lung cancer advanced disease, samples available for molecular diagnosis are very limited, and usually, it is not possible to perform several gene analysis separately. In addition, it would be more time-consuming and, eventually, equally or even more expensive to design primers for all hotspot alterations needed.

The raw data will open source?

A comprehensive list of the most relevant results will be available as supplementary material. The raw datasets used and analyzed during the current study will be available from the corresponding author on a reasonable request.

Reviewer 2 Report

Authors tested NGS-based approach and Sanger sequencing plus FISH approach to detect actionable somatic mutations in lung adenocarcinoma. The results of each platform were compared in an experimental cohort and an implementation cohort. Mutations in EGFR demonstrated high concordance between NGS and Sanger sequencing. A combined DNA and RNA NGS panel identified more patients harboring targetable genetic alterations.

Although this study used sound methods and validated data, there are some points to be addressed besides the novelty of this study.   

 1. Authors used several platforms to detect genetic alterations including Sanger sequencing, FISH for ALK translocation, target NGS, and DNA plus RNA NGS panel. The sensitivity of each method according to specimen types or mutation types could be compared.

2. According to figure 3, the patients harboring EGFR mutations showed the most favorable OS. Among patients with actionable mutations, 30 were treated with targeted therapy. Are those patients were treated with EGFR TKI? More information about targeted therapy and detected mutations will be helpful to understand prognosis of those patients.  

3. Figure legends should include more detailed description.

Author Response

Authors used several platforms to detect genetic alterations including Sanger sequencing, FISH for ALK translocation, target NGS, and DNA plus RNA NGS panel. The sensitivity of each method according to specimen types or mutation types could be compared.

We thank the reviewer for highlighting this aspect. In fact, that analysis was performed. However, we decided not included because it would divert the message and increase the manuscript dimension. Considering the large variety of samples analyzed, which we consider a strength of the study, sensitivity was calculated by the type of sample (histological versus cytological). In the validation cohort, the purpose was to validate and compare NGS with the standard method at the time. We did not find different frequencies of alterations in the two group of samples, histological versus cytological (7/21 mutations in cytological samples, vs 49/96 in histological samples; P=0.141). In the implementation cohort, sequencing was successful in both cytological and histological samples, with no statistical difference in the sequencing rate (16/18 for cytology and 102/105 for histology, P=0.1). Hotspot alterations found in both types of samples, although in higher percentage in histological samples (12/18, 66/105, respectively, P=0.538).

According to figure 3, the patients harboring EGFR mutations showed the most favorable OS. Among patients with actionable mutations, 30 were treated with targeted therapy. Are those patients were treated with EGFR TKI? More information about targeted therapy and detected mutations will be helpful to understand prognosis of those patients.

Among the 30 patients with “targetable” alterations 21/23 EGFR, 2/5 BRAF, 7/9 (ALK) were treated with TKIs. As groups were small, no statistical comparisons were feasible. Treatment with TKIs can improve patient´s survival and for that reason, one key clinical concern is to detect all candidates to TKI´s treatment.

Figure legends should include more detailed description.

We thank the reviewer. Additional information with more detail has been included in the figure legends.

Reviewer 3 Report

The manuscript by Fernandes et al compares the clinical application of panel NGS testing for “hotspot” mutations in lung cancer with the current practice of sequential Sanger sequencing for individual mutations.  The data show a clear improvement in the ability to determine the presence of pathogenic and, in some instances, actionable mutations.  The manuscript is clearly written and the overall presentation is satisfactory, if a bit visually cumbersome.

My main critique is that this is a rapidly evolving space in translational cancer care and the limitations of the panel sequencing approach, and indeed the specific panel used, are inadequately addressed in the discussion.  A specific concern is the very low rate of detection of alterations in tumour suppressors Tp53 and STK11 (compared for example with the TCGA deep sequencing rates of detection).  The panel used appears to offer only low coverage over a relatively small number of genes, and broader panels are readily available on the market (I do accept that this may not have been the case from the outset of the study).  A very large number of cases remained unclassifiable even after panel NGS.

A final point relates to use of the term “non-mutated” to describe cases where no mutation was detected, almost certainly due to the limitations of the approach used.  It would be better to refer to such cases as “unclassifiable” under the methodology employed.

Author Response

My main critique is that this is a rapidly evolving space in translational cancer care and the limitations of the panel sequencing approach, and indeed the specific panel used, are inadequately addressed in the discussion. A specific concern is the very low rate of detection of alterations in tumour suppressors Tp53 and STK11 (compared for example with the TCGA deep sequencing rates of detection). The panel used appears to offer only low coverage over a relatively small number of genes, and broader panels are readily available on the market (I do accept that this may not have been the case from the outset of the study). A very large number of cases remained unclassifiable even after panel NGS.

We thank the reviewer for this important topic. We completely agree that this is a rapidly evolving space in the translational cancer care with new biomarkers added to guidelines on a yearly basis. This is in fact the reason why, in the manuscript, different NGS panels were used along the time the study took place. Additionally, in the context of lung cancer and on the clinical standpoint, our primary concern was to cover genes with targetable drugs, already approved, for off-label use or on clinical trials. Our goal was not a discovery panel; although we recognize that a broader panel or even whole-exome could contribute to a better understand the disease, especially in the context of progressive disease where more complex alterations can occur. At this moment, clinically targetable alterations are covered with this panel with a better cost/efficacy ratio. In addition, shorter targeted gene panels have higher sensitivity than larger panels (more than 300 genes) or exomes, enabling the detection of low-frequency mutations even when low-input of tumor DNA are available, as it was the case of this study. The strategy used resulted in a percentage of unclassified patients similar to other published studies (PMID:28780976).

A final point relates to use of the term “non-mutated” to describe cases where no mutation was detected, almost certainly due to the limitations of the approach used. It would be better to refer to such cases as “unclassifiable” under the methodology employed.

We agree and thank the reviewer for this comment. We have changed accordingly the terminology throughout the manuscript.